# FedTAP: Federated Multi-Task Continual Learning via Dynamic Task-Aware Prototypes

## Abstract

Real-world federated learning systems often involve clients performing different tasks under continually changing conditions, including dynamic participation, where new tasks emerge while others fade away. However, this dynamic environment presents complexity beyond the scope of conventional approaches. Particularly, Federated Multi-Task Learning address different tasks but assumes they are static, while Federated Continual Learning only considers temporal data shifts within a single-task. To address this gap, we introduce a **F**ederated **M**ulti-**T**ask **C**ontinual **L**earning (**FMTCL**), a novel scenario that simultaneously handles task heterogeneity, temporal data shifts, and dynamic task composition. We propose **FedTAP**, **Fed**erated **T**ask-**A**ware **P**rototype, a prototype-based framework designed to solve the challenges of FMTCL. It consists of: *(i)* Prototype-Guided Aggregation (PGA), which aggregates client updates in a shared prototype space, *(ii)* Task-Aware Prototype Learning (TPL), which trains a diverse and sparsely utilized set of prototypes, and *(iii)* Adaptive Prototype Allocation (APA), which manages the prototype pool to adapt a dynamic task participation. FedTAP achieves state-of-the-art performance on multi-task benchmarks, demonstrating strong effectiveness in FMTCL.

## 1 Introduction

Federated Learning (FL) (McMahan et al., 2017; Konečnỳ et al., 2015; Kairouz et al., 2021) enables collaborative model training across decentralized clients without sharing raw data, making it particularly suitable for privacy-sensitive domains such as healthcare and industry (Bercea et al., 2021; Dayan et al., 2021; Warnat-Herresthal et al., 2021; Zhang et al., 2022; Yang et al., 2019). By aggregating locally trained models across multiple clients, FL enables each client to benefit from a global model knowledge derived from more diverse data than its own data, resulting in a model that outperforms those trained in isolation (Zhao et al., 2018; Li et al., 2020). However, in practice, this process faces three interconnected challenges: *task heterogeneity* where clients perform different tasks, such as segmentation or classification, *temporal data shift* where a client's data distribution changes over time, and *dynamics* as new tasks emerge and some leave. These challenges must be considered together, given their intricate interplay, as dynamic composition determines the active set of heterogeneous tasks at any given round, and temporal shifts concurrently modify the data distributions of those tasks.

Unfortunately, existing approaches only address part of these challenges. Specifically, Federated Multi-Task Learning (FMTL) (Smith et al., 2017; Cai et al., 2023; Chen et al., 2024; Yang et al., 2024; Lu et al., 2024) primarily tackles task heterogeneity by decomposing models into task-specific parameters. But FMTL methods assume that the set of participating tasks remains unchanged across all rounds, making it difficult to handle scenarios where the task set changes over time. On the other hand, to handle *temporal data shift*, Federated Continual Learning (FCL) (Guo et al., 2024; Qi et al., 2023; Tran et al., 2024; Dong et al., 2022; Liang et al., 2024; Piao et al., 2024) has been studied to adapt global models to sequentially changing data while retaining the previously learned knowledge, typically

through weight-level regularization to constrain parameter updates. Yet FCL assumes the same task across clients, which does not hold in task heterogeneity setting. Moreover, since FCL assumes a static task participation across rounds, it cannot handle dynamic task participation. Thus, while FMTL and FCL each address one aspect of the three interconnected challenges, neither can fully resolve them simultaneously.

To overcome all these challenges, we propose **Federated Multi-Task Continual Learning (FMTCL)**: a novel federated learning scenario that simultaneously addresses task heterogeneity across clients, temporal data shift within each task, and dynamic task composition.

Table 1: **Comparison of coverage in federated learning settings.** FMTL addresses only task heterogeneity, while FCL focuses solely on temporal data dynamics. FMTCL tackles all three challenges: task heterogeneity, temporal data shift, and dynamic task composition.

| Challenges | FMTL | FCL | **FMTCL (Ours)** |
|---|---|---|---|
| Task heterogeneity | ✓ | ✗ | ✓ |
| Temporal data shift | ✗ | ✓ | ✓ |
| Dynamic task composition | ✗ | ✗ | ✓ |

Table 1 summarizes the differences between existing FL scenarios, FMTL and FCL, and our proposed scenario, FMTCL. This setting is especially relevant in real-world applications like healthcare and manufacturing, where client roles are specialized and their participation changes over time. In healthcare, hospitals comprise specialized departments, each responsible for unique diagnostic tasks. For example, respiratory hospitals specialized in infectious diseases join the federation during a pandemic, and withdraw once the situation stabilizes, while mental health institutions join later as psychological concerns rise. The global federation must thus adapt to changing task composition while maintaining diagnostic performance. Similarly, in manufacturing, each production line specializes in tasks such as semiconductor processing or packaging, requiring management of dynamic composition as new lines are added and outdated ones removed.

A straightforward approach to tackle FMTCL is to combine techniques from FMTL and FCL. However, this naive integration is insufficient and leads to two critical limitations: *(i) Conflict between learning objectives: adaptation and stability.* FMTL enables task-specific adaptation, a model's ability to learn from heterogeneous tasks, by utilizing a shared backbone and separate task-specific parameters, whereas FCL enforces stability, a model's ability to preserve knowledge over time, by constraining important parameters. When combined, FMTL's requirement to change the shared parameters to find a compromise for multiple new tasks directly opposes FCL's requirement to preserve those same shared parameters to retain knowledge from old tasks. Even assigning a separate FCL model to each task not only incurs substantial memory and communication overhead, but also breaks the core principle of continual learning, as each model handles a single task in isolation without a shared global model which is an essential characteristic of federated learning. *(ii) Representational capacity misallocation in dynamic tasks.* Without an adaptive mechanism for dynamic task composition, which neither FMTL nor FCL provides, task-specific parameters from non-participating tasks remain in the global model. When a new, unrelated task arrives, a naive system forces it to start learning from this residual knowledge, a specialized but often irrelevant starting point that causes negative transfer. This process hinders the learning of new, task-specific features and ultimately degrades performance on current tasks.

To address these two challenges, we propose a novel method, **FedTAP** (**Fed**erated **T**ask-**A**ware **P**rototype), which consists of three components: Prototype-Guided Aggregation (PGA), Task-Aware Prototype Learning (TPL), and Adaptive Prototype Allocation (APA). To mitigate conflict between adaptation and stability, FedTAP shifts from direct parameter updates in conventional FL to an indirect, representation-based approach. To this end, (i) PGA enables task-specific adaptation by translating each client update into a unique combination of shared prototypes, and preserves stability, as the shared prototype base is not directly altered by any single update. Moreover, (ii) TPL supports this by learning diverse prototypes and enforcing sparse attention, keeping each task's combination compact. To address representational capacity misallocation in dynamic tasks, (iii) APA dynamically expands the prototype pool with new prototypes and prunes obsolete ones, ensuring the model's representational resources are efficiently allocated to current tasks.

Our main contributions are summarized as follows:

- For the first time, we propose Federated Multi-Task Continual Learning (**FMTCL**), a realistic federated learning scenario that reflects three key challenges, including heterogeneous tasks, temporal data shift, and dynamic task composition.

- We propose Federated Task-Aware Prototype (**FedTAP**), a novel prototype-based framework to address the challenges of FMTCL. It mitigates the conflict between adaptation and stability through Prototype-Guided Aggregation (PGA), which calculates client updates as sparse combinations of the shared prototype bases learned through Task-Aware Prototype Learning (TPL).

- We further address capacity misallocation under dynamic task composition through Adaptive Prototype Allocation (APA), which dynamically manages the prototype pool to control the alignment of model capacity with evolving tasks.

- We conduct extensive experiments on multi-task benchmarks under diverse dynamic scenarios. Experimental results demonstrate that FedTAP consistently outperforms existing baselines in both generalization and task-specific performance, validating its effectiveness in the proposed FMTCL.

## 2 RELATED WORK

**Federated Multi-Task Learning.** Federated Multi-Task Learning (FMTL) enables clients to collaboratively train models while each works on distinct tasks. Early work like MOCHA (Smith et al., 2017) proposed learning separate models per client with shared knowledge across them, but assumed similar tasks—referring to variations in class distributions rather than fundamentally different task types. FedBone (Chen et al., 2024) improved performance by sharing encoders from different clients, although each client was still limited to one task. MAS (Zhuang et al., 2023) supported multi-task clients, but only allowed collaboration between clients with exactly the same task sets. MaT-FL (Cai et al., 2023) introduced a more practical scenario where each client performs a unique task and used dynamic grouping to reduce conflicts during model updates, making it one of the first model that address task heterogeneity. More recently, FedHCA$^2$ (Lu et al., 2024) proposed the Hetero-Client FMTL setting, which supports clients with different numbers and types of tasks by splitting the model into encoders and decoders and applying a cross-attention mechanism to combine task-specific knowledge more effectively. While these methods help address task diversity, they assume that the set of tasks remains fixed over time.

**Federated Continual Learning.** Federated Continual Learning (FCL) requires a global model to adapt continually as client data and class distributions change over time. Regularization-based methods (Yoon et al., 2021; Luo et al., 2023), leverage parameter importance metrics like fisher information to constrain updates on weights critical to previously learned classes, thereby mitigating catastrophic forgetting. Replay-based approaches (Qi et al., 2023; Babakniya et al., 2023) use local generators to recreate past data distributions. In knowledge-distillation methods (Ma et al., 2022) a fixed teacher model provides soft labels or feature hints to the student model, guiding it to retain knowledge of previously learned classes. More recently, LANDER (Tran et al., 2024) has demonstrated that anchoring synthetic sample generation on pretrained label embeddings. FedTA (Yu et al., 2025) tackles spatial-temporal heterogeneity in FCL by introducing trainable tail anchors and prototype selection to alleviate catastrophic forgetting. Although these advances mitigate catastrophic forgetting and distribution shift, most FCL methods still assume a static class set and face challenges when new classes (tasks) continuously arrive.

## 3 METHOD

In this section, we introduce FedTAP, a prototype-based framework for Federated Multi-Task Continual Learning (FMTCL). In Section 3.1, we formally define the FMTCL setting, including the participation types of tasks. Section 3.2 presents the overall federated learning procedure of FedTAP. Section 3.3, Section 3.4, and Section 3.5 describe the three components

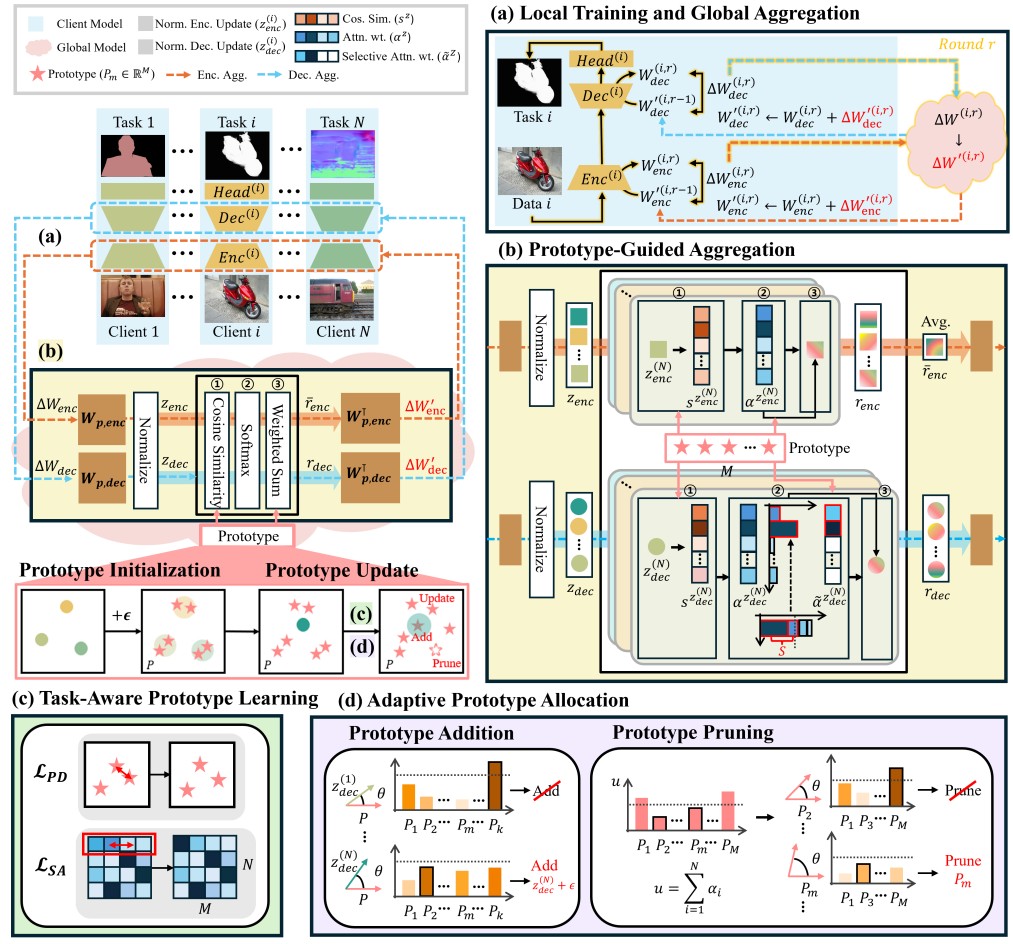

Figure 1: **Overview of FedTAP.** (a) Each client performs local training with its own encoder and decoder, and sends updates to the global model. (b) Through PGA, the global model aggregates both encoder-side and decoder-side updates from clients using the prototypes updated by TPL and APA. (c) TPL adjusts the prototypes using two complementary losses. (d) APA dynamically manages the prototype pool through addition and pruning.

of FedTAP: Prototype-Guided Aggregation (PGA), Task-Aware-Prototype Learning (TPL) and Adaptive Prototype Allocation (APA), respectively.

## 3.1 PROBLEM SETUP: FEDERATED MULTI-TASK CONTINUAL LEARNING

Federated Multi-Task Continual Learning (FMTCL) is a novel federated learning scenario where clients perform different tasks, with local data distributions shift over rounds per clients, and task participation changes dynamically. Given the overall client set $C = \{c_1, c_2, \ldots, c_N\}$, each client $c_i \in C$ is assigned to a unique task $T_i$ and holds a dataset $D_i^r$ that changes over rounds $R = \{1, 2, \ldots, r, \ldots\}$ due to temporal shift in data. In each round $r$, only a subset of clients $C^{(r)} \subseteq C$ actively participates. The task pool at round $r$ is then defined as $T^{(r)} = \{T_i \mid c_i \in C^{(r)}\}$. As the participating clients differ across rounds, the task pool changes accordingly. These changes result in four types of tasks over time: (i) *active tasks*, whose clients continue to participate, (ii) *inactive tasks*, whose clients are temporarily absent but may return, (iii) *new tasks*, introduced by newly joined clients, and (iv) *left tasks*, whose clients have been absent for a long time and unlikely to return.

We follow the standard federated learning protocol, where each client model trains locally and sends its updated parameters to the global model. The global model aggregates updates from participating clients and returns the aggregated result to each client. This process is

repeated over multiple communication rounds, progressively training the global model in a way that ultimately improves the performance of client models on their respective tasks.

## 3.2 Local Training and Global Aggregation

Each client independently performs local training using its assigned task and data. The $i$-th client has an encoder $\text{Enc}^{(i)}$, initialized from a common global model across clients, and a task-specific decoder $\text{Dec}^{(i)}$, both trained on local data $D_i$. While the encoder architectures are shared across clients, the parameters are maintained and updated locally, resulting in independent encoders across clients. As illustrated in Figure 1(a), which describes the local training and global aggregation process at round $r$, each client begins with $W_{\text{enc}}^{\prime(i,r-1)}$ and $W_{\text{dec}}^{\prime(i,r-1)}$ and updates them to $W_{\text{enc}}^{(i,r)}$ and $W_{\text{dec}}^{(i,r)}$ through local training. Then local updates are computed as their difference, as follows:

$$\Delta W_{\text{enc}}^{(i,r)} = W_{\text{enc}}^{(i,r)} - W_{\text{enc}}^{\prime(i,r-1)}, \quad \Delta W_{\text{dec}}^{(i,r)} = W_{\text{dec}}^{(i,r)} - W_{\text{dec}}^{\prime(i,r-1)}. \tag{1}$$

These updates $\Delta W_{\text{enc}}^{(i,r)}$ and $\Delta W_{\text{dec}}^{(i,r)}$ are transmitted to the global model. Therefore, the $i$-th client performs two separate parameter updates: one for the encoder, which contributes to task-agnostic representation learning, and another for the decoder, which captures the task-specific one. The global model aggregates the received updates from all participating clients to compute global updates $\Delta W_{\text{enc}}^{\prime(i,r)}$ and $\Delta W_{\text{dec}}^{\prime(i,r)}$. (see Sec. 3.3 for details.) These updates are then transmitted to each client, which applies them to its personalized parameters as:

$$W_{\text{enc}}^{\prime(i,r)} \leftarrow W_{\text{enc}}^{(i,r)} + \Delta W_{\text{enc}}^{\prime(i,r)}, \quad W_{\text{dec}}^{\prime(i,r)} \leftarrow W_{\text{dec}}^{(i,r)} + \Delta W_{\text{dec}}^{\prime(i,r)}. \tag{2}$$

These updated parameters $W_{\text{enc}}^{\prime(i,r)}$ and $W_{\text{dec}}^{\prime(i,r)}$ are then used to initialize the client's model for the next round $r+1$.

## 3.3 Prototype-Guided Aggregation

In this section, we describe Prototype-Guided Aggregation (PGA) (see Figure 1(b)), the core mechanism that reduces interference among heterogeneous tasks by aligning client updates with a prototype space. Instead of directly aggregating the raw client updates, we use a shared set of $M$ prototypes $P = \{p_1, p_2, \ldots, p_m, \ldots, p_M\}$ that are derived from task-specific decoder updates but used across both encoder and decoder updates during aggregation. Encoder updates are inherently task-agnostic, and simple averaging would ignore task-specific signals and cause instability. To address this, encoder updates are also aligned with task-aware prototypes derived from decoder updates. The mapping of both encoder and decoder updates into the prototype space is performed through two trainable projection matrices, $W_{p,\text{enc}}$ and $W_{p,\text{dec}}$, which are updated jointly with the prototypes. This alignment prevents encoder and decoder updates from drifting apart and enables global aggregation to combine encoder-based shared representations with decoder-guided task-specific signals. The details of prototype construction and learning are described in the following section.

**Encoder-side Aggregation in Global Model.** Each client's encoder update $\Delta W_{\text{enc}}^{(i)}$ is first projected into the prototype space using $W_{p,\text{enc}}$ and then $\ell_2$-normalized:

$$z_{\text{enc}}^{(i)} = \text{Normalize}\left(W_{p,\text{enc}} \cdot \Delta W_{\text{enc}}^{(i)}\right). \tag{3}$$

We compute attention weights by applying a Softmax function to the cosine similarities between $z_{\text{enc}}^{(i)}$ and each prototype $P_m$ (refer to Eq. 4):

$$\alpha_m^{z_{\text{enc}}^{(i)}} = \text{Softmax}\left(s_m^{z_{\text{enc}}^{(i)}}\right), \quad \text{where} \quad s_m^{z_{\text{enc}}^{(i)}} = \cos(z_{\text{enc}}^{(i)}, P_m). \tag{4}$$

Using $\alpha_m^{z_{\text{enc}}^{(i)}}$, an intermediate prototype-guided representation $(r_{\text{enc}}^{(i)})$ is computed as follows:

$$r_{\text{enc}}^{(i)} = \sum_{m=1}^{M} \alpha_m^{z_{\text{enc}}^{(i)}} \cdot P_m. \tag{5}$$

We allow all prototypes to contribute without applying sparsity, so that the encoder update reflects a broad combination of prototype directions. This encourages the encoder to learn generalizable patterns that are useful across tasks. We then average all client intermediate prototype-guided representations and project the result back to the original parameter dimensions using the transposed projection matrix as:

$$\Delta W_{\text{enc}}'^{(i)} = W_{p,enc}^\top \cdot \tilde{r}_{\text{enc}}, \quad \text{where} \quad \tilde{r}_{\text{enc}} = \frac{1}{N} \sum_{i=1}^{N} r_{\text{enc}}^{(i)}. \tag{6}$$

**Decoder-side Aggregation in Global Model.** Similar to the encoder-side process, the decoder update $\Delta W_{\text{dec}}^{(i)}$ is projected into the prototype space using $W_{p,\text{dec}}$ and normalized:

$$z_{\text{dec}}^{(i)} = \text{Normalize}\left(W_{p,\text{dec}} \cdot \Delta W_{\text{dec}}^{(i)}\right). \tag{7}$$

Unlike the encoder, we apply a dynamic sparsity mechanism to select a relevant subset of prototypes for each task. This selection is guided by an initial attention distribution, $\alpha_m^{z_{\text{dec}}^{(i)}}$, computed via Softmax over the cosine similarities between $z_{\text{dec}}^{(i)}$ and each prototype $P_m$. The subset $S_i$ is then formed by accumulating the prototypes with the highest attention weights until their cumulative weight surpasses a predefined threshold $\tau_k$:

$$S_i = \{m_1, \ldots, m_{k_i}\} \quad \text{s.t.} \quad \sum_{j=1}^{k_i} \alpha_{m_j}^{z_{\text{dec}}^{(i)}} \geq \tau_k \quad \text{and} \quad \sum_{j=1}^{k_i-1} \alpha_{m_j}^{z_{\text{dec}}^{(i)}} < \tau_k, \tag{8}$$

where $\{m_j\}$ are sorted by their weights in $\alpha_{\text{dec}}^{(i)}$. The final sparse attention weights, $\tilde{\alpha}_m^{z_{\text{dec}}^{(i)}}$, are computed by applying Softmax to the cosine similarities of only the prototypes in the subset $S_i$. The final decoder update is the average of the original update and the prototype-guided representation ($r_{\text{dec}}$) which is projected back to the original parameter space using $W_{p,dec}^\top$:

$$\Delta W_{\text{dec}}'^{(i)} = \frac{1}{2}\left(\Delta W_{\text{dec}}^{(i)} + W_{p,\text{dec}}^\top \cdot r_{\text{dec}}^{(i)}\right), \quad \text{where} \quad r_{\text{dec}}^{(i)} = \sum_{m \in S_i} \tilde{\alpha}_m^{z_{\text{dec}}^{(i)}} P_m. \tag{9}$$

### 3.4 Task-Aware Prototype Learning

To address the conflict between : adaptation and stability in FMTCL, we propose a training strategy, called Task-Aware Prototype Learning (TPL), which leverages a set of shared prototypes from decoder-side updates containing task-specific signals (see Figure 1(c)).

**Prototype Initialization.** The initial prototypes are generated by adding small Gaussian noise $\epsilon \sim \mathcal{N}(0,1)$ to normalized decoder updates $z_{\text{dec}}^{(i)}$ (as defined in Eq 7), collected during the first round. Rather than using purely random initialization, this approach leverages meaningful task-specific update directions while injecting stochastic variation, which prevents early collapse into the same direction and encourages broader exploration.

**Loss Functions.** To ensure a stable and task-aware representation, our prototype learning is guided by two core, complementary loss functions.

First, the Prototype Diversity (PD) loss, $\mathcal{L}_{\text{PD}}$, enforces orthogonality among prototypes. This loss ensures that prototypes span distinct directions, reducing redundancy and forming an expressive representational basis. The equation is formulated as follows:

$$\mathcal{L}_{\text{PD}} = \left\| PP^\top - I \right\|_F^2. \tag{10}$$

Second, the Sparse Attention (SA) loss, $\mathcal{L}_{\text{SA}}$, encourages each client to focus its attention on a small number of prototypes. This loss is formulated as the entropy of the final sparse

attention distribution, which is minimized to sharpen the focus. The equation is as follows:

$$\mathcal{L}_{\text{SA}} = \sum_{i=1}^{N} \sum_{m \in S_i} \tilde{\alpha}_m^{z_{\text{dec}}^{(i)}} \log \tilde{\alpha}_m^{z_{\text{dec}}^{(i)}}. \tag{11}$$

By minimizing this entropy, the model guides each task to select only a few meaningful prototypes, improving both interpretability and efficiency by reducing reliance on redundant or irrelevant prototype combinations.

We combine these two objectives into a single prototype-level loss, called $\mathcal{L}_{\text{proto}}$. While $\mathcal{L}_{\text{PD}}$ imposes a constraint that enforces mutual orthogonality among all prototypes, $\mathcal{L}_{\text{SA}}$ leverages each client's update direction to encourage a sparse selection of prototypes, thereby aligning a relevant subset with each task. Therefore, the final loss function is constructed as follows:

$$\mathcal{L}_{\text{proto}} = \mathcal{L}_{\text{PD}} + \mathcal{L}_{\text{SA}}. \tag{12}$$

### 3.5 ADAPTIVE PROTOTYPE ALLOCATION

To ensure that the prototype pool adapts to the dynamic task composition, Adaptive Prototype Allocation (APA) dynamically adds new prototypes and prunes obsolete ones (see Figure 1(d)). The decision to add is guided by how well client updates are represented by the existing pool, while the decision to prune is based on the usage and isolation of each prototype.

**Prototype Addition.** To expand the expressiveness of the prototype set, a new prototype is introduced when a client's update is not well-represented by any existing prototypes. Specifically, this is determined by checking whether the maximum cosine similarity between the client's projected update $z_{\text{dec}}^{(i)}$ and all prototypes is below a predefined threshold $\delta_{\text{add}}$. Clients that satisfy this condition are grouped into a set $U$ as:

$$U = \{ i \mid \max_m \cos(z_{\text{dec}}^{(i)}, P_m) < \delta_{\text{add}} \}. \tag{13}$$

Their projected updates are averaged to form the new prototype $P_{\text{new}}$:

$$P_{\text{new}} = \text{Normalize} \left( \frac{1}{|U|} \sum_{i \in U} z_{\text{dec}}^{(i)} + \epsilon \right), \tag{14}$$

where a small Gaussian noise $\epsilon \sim \mathcal{N}(0, I)$ is added to maintain diversity.

**Prototype Pruning.** To maintain a compact and efficient pool, we prune prototypes based on two criteria: their usage score across clients and their representational isolation score from other prototypes. The usage score of each prototype $P_m$ is the total initial attention it receives from all clients, $u_m = \sum_{i=1}^{N} \alpha_{dec,m}^{(i)}$. These scores are then normalized across all prototypes to enable consistent thresholding across rounds ($\tilde{u}_m$). The isolation score of each prototype $P_m$ is the maximum cosine similarity with other prototypes, $i_m = \max_{j \neq m} \cos(P_m, P_j)$. A prototype $P_m$ is pruned if it satisfies both of the following conditions:

$$\tilde{u}_m < \frac{1}{M} \quad \text{and} \quad i_m < \delta_{\text{prune, sim}}, \quad \text{where} \quad \tilde{u}_m = \frac{u_m}{\sum_{m'=1}^{M} u_{m'}}. \tag{15}$$

Here, $M$ is the total number of prototypes and $\delta_{\text{prune, sim}}$ is adaptively set as $\mu - \sigma$, where $\mu$ and $\sigma$ denote the mean and standard deviation of all pairwise prototype similarities. Using both conditions together prevents unnecessary pruning of prototypes that are infrequently used but still semantically connected, or conversely those that are isolated but still play an important role. As a result, through APA, the model efficiently allocates its capacity to informative and frequently used components.

Table 2: **Main Table.** FedTAP achieves the best performance across all five tasks in the FMTCL.

| Setting | Method | Task / Task Status | SemSeg (mIoU ↑) Inactive | Parts (mIoU ↑) Left | Sal (maxF ↑) Active | Normals (100-mErr ↑) New | Edge (odsF ↑) Inactive | Avg. (↑) |
|---|---|---|---|---|---|---|---|---|
| | Local | | 17.22 | 26.14 | 80.05 | 79.71 | 56.02 | 51.82 |
| Traditional FL | FedAvg (McMahan et al., 2017) | | 16.03 | 27.26 | 81.96 | 80.66 | 55.04 | 52.19 |
| | FedProx (Li et al., 2020) | | 17.03 | 26.54 | 80.56 | 79.56 | 55.67 | 51.87 |
| | Ditto (Li et al., 2021) | | 17.09 | 27.52 | 82.04 | 80.63 | 57.03 | 52.86 |
| FCL | LANDER Tran et al. (2024) | | 17.70 | 28.78 | 81.11 | 80.27 | 56.88 | 52.94 |
| | FedTA (Yu et al., 2025) | | 18.03 | 29.32 | 81.91 | 79.94 | 56.21 | 53.08 |
| FMTL | MOCHA (Smith et al., 2017) | | 17.74 | 28.95 | 81.29 | 81.88 | 56.38 | 53.24 |
| | MaT-FL (Cai et al., 2023) | | 18.92 | 29.12 | 82.37 | 81.37 | 56.81 | 53.71 |
| | FedHCA$^2$ (Lu et al., 2024) | | 19.26 | 28.51 | 84.28 | 80.18 | 57.95 | 54.03 |
| | **FedTAP (Ours)** | | **23.11** | **33.90** | **86.12** | **82.75** | **61.13** | **57.40** |

Table 3: **Ablation on TPL and APA.** Disabling either TPL or APA results in a notable performance degradation, confirming the contribution of both components to the overall effectiveness of FedTAP.

| TPL | APA | Active | Inactive | Left | New | Avg. |
|---|---|---|---|---|---|---|
| ✗ | ✗ | 79.83 | 71.06 | 25.07 | 77.94 | 50.78 |
| ✓ | ✗ | 84.09 | 78.45 | 30.05 | 80.75 | 54.66 |
| ✗ | ✓ | 82.13 | 73.40 | 27.94 | 79.72 | 52.63 |
| ✓ | ✓ | **86.12** | **84.24** | **33.90** | **82.75** | **57.40** |

Table 4: **Ablation on Prototype Sharing Strategy.** Utilizing a unified prototype space for both encoder and decoder updates enhances knowledge sharing and leads to superior performance.

| $P$ | Active | Inactive | Left | New | Avg. |
|---|---|---|---|---|---|
| Separate | 84.96 | 80.28 | 30.89 | 81.27 | 55.48 |
| **Shared** | **86.12** | **84.24** | **33.90** | **82.75** | **57.40** |

## 4 EXPERIMENTS

### 4.1 EXPERIMENTAL SETUP

We use the PASCAL-Context (Mottaghi et al., 2014) with five tasks: semantic segmentation (SemSeg), human parts segmentation (Parts), saliency detection (Sal), surface normal estimation (Normals), and edge detection (Edge). Each client performs local training on task-specific data, with an equal number of randomly sampled data at each round. For evaluation, we follow standard metrics: mean Intersection over Union (mIoU) for SemSeg and Parts, maximum F-measure (maxF) for Sal, mean angular error (mErr) for Normals, and optimal dataset scale F-measure (odsF) for Edge. We compare our method with established baselines across three categories: traditional federated learning (FedAvg (McMahan et al., 2017), FedProx (Li et al., 2020), Ditto (Li et al., 2021)), federated continual learning (LANDER (Tran et al., 2024), FedTA (Yu et al., 2025)), and federated multi-task learning (MOCHA (Smith et al., 2017), MaT-FL (Cai et al., 2023), FedHCA$^2$ (Lu et al., 2024)). Details of models and parameters are provided in the supplementary material (Sec. A.2.1).

### 4.2 MAIN RESULTS

To configure the FMTCL, tasks are categorized as active (Sal, Edge), inactive (SemSeg), left (Parts), and new (Normals), based on their correlation across tasks. Details of the scenarios, including the exact participation rounds for each task type, are provided in the supplementary material (Sec. A.2.2). Performance is measured on the local model at the final round each task participates in. For example, inactive tasks (participating in rounds 1-15 and 35-50) are evaluated at round 50. To ensure higher values consistently indicate better performance across all metrics, for surface normals, we report $100 - mErr$. As shown in Table 2, FedTAP achieves the best overall performance, outperforming all baselines across the five tasks. Additional results for various task combinations are included in the supplementary material (Sec. C.3).

Table 5: **Ablation of TPL Loss Functions.** Removing either $\mathcal{L}_{PD}$ or $\mathcal{L}_{SA}$ degrades performance, confirming that both are essential for learning an effective set of prototypes.

| $\mathcal{L}_{PD}$ | $\mathcal{L}_{SA}$ | Active | Inactive | Left | New | Avg. |
|---|---|---|---|---|---|---|
| ✗ | ✗ | 82.13 | 73.40 | 27.94 | 79.72 | 52.63 |
| ✓ | ✗ | 83.95 | 76.39 | 28.55 | 80.4 | 53.85 |
| ✗ | ✓ | 82.58 | 75.44 | 27.81 | 79.93 | 53.15 |
| ✓ | ✓ | **86.12** | **84.24** | **33.90** | **82.75** | **57.40** |

Table 6: **Ablation of APA Mechanisms.** The results show that both prototype addition for new tasks and pruning for obsolete ones are essential for the dynamic adaptation performed by APA.

| Addition | Pruning | Active | Inactive | Left | New | Avg. |
|---|---|---|---|---|---|---|
| ✗ | ✗ | 84.09 | 78.45 | 30.05 | 80.75 | 54.66 |
| ✓ | ✗ | 85.30 | 81.05 | 32.22 | 81.34 | 55.98 |
| ✗ | ✓ | 84.13 | 79.44 | 30.13 | 80.98 | 54.93 |
| ✓ | ✓ | **86.12** | **84.24** | **33.90** | **82.75** | **57.40** |

### 4.3 ABLATION STUDIES

The detailed ablation results show the necessity of each design component in FedTAP. As shown in Table 3, removing either the prototype learning (TPL) or adaptation (APA) function leads to a significant performance degradation. Table 4 demonstrates the superiority of a shared prototype space, which creates synergy between the general representations of the encoders and the task-specific knowledge of the decoders. Furthermore, Table 5 indicates that performance drops in the absence of either the diversity or sparsity, showing the need of a balance between expanding knowledge and specifying its application. Finally, Table 6 shows that optimal performance requires balancing the acquisition of new knowledge through prototype addition and the pruning of obsolete knowledge. Further ablation results on the PGA strategies are provided in the supplementary material (Sec. B.1).

### 4.4 ANALYSIS

Figure 2 demonstrates FedTAP's ability to retain knowledge under dynamic task composition, particularly for inactive tasks. When inactive tasks reparticipate at round 36 after an absence, FedHCA$^2$ suffers from catastrophic forgetting, whereas FedTAP preserves the acquired knowledge and resumes training from a higher score. This retention is enabled by the re-use of prototypes, as evidenced by the absence of a sudden increase in the prototype count when the tasks reappears. Such preservation and re-use knowledge highlight an advantage of FedTAP in the FMTCL setting. Further analysis on prototype space is given in the supplementary material (Sec. B.2).

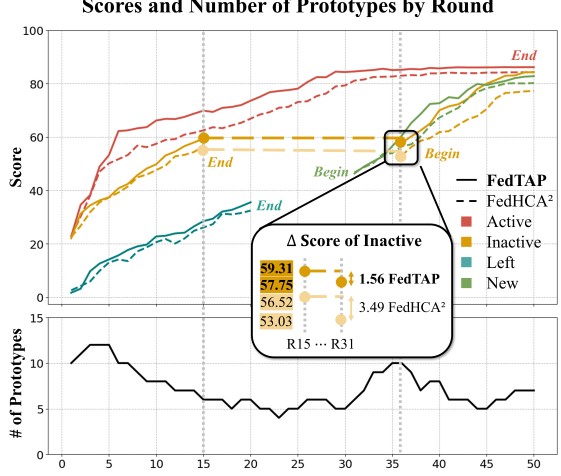

Figure 2: **Scores and Number of Prototypes per Round.** The results illustrates the link between adaptative prototype management via APA (bottom) and its resulting performance (top). FedTAP generally outperforms FedHCA$^2$ across all task types. Red, yellow, blue, and green lines correspond to Active, Inactive, Left, and New tasks, respectively. Solid lines represent FedTAP, while dashed lines indicate FedHCA$^2$.

## 5 CONCLUSION

In this work, we introduced Federated Multi-Task Continual Learning (FMTCL), a novel federated learning scenario addressing the combined challenges of task heterogeneity, temporal data shift, and dynamic task composition, for which existing methods like FMTL and FCL are insufficient. We proposed FedTAP (Federated Task-Aware Prototype), a novel prototype-based framework that resolves learning objective conflicts through Prototype-Guided Aggregation (PGA) in a shared space of prototypes, which are learned to be diverse and used sparsely through Task-Aware Prototype Learning (TPL) and dynamically managed by Adaptive Prototype Allocation (APA) to prevent capacity misallocation in dynamic task. Experimental results demonstrate that FedTAP outperforms existing methods across tasks, validating its effectiveness in FMTCL setting. We hope our work provides a new research direction toward realistic federated learning in dynamic environments.

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

# A ADDITIONAL INFORMATION

## A.1 NOTATION

Table 7: **Summary of Notation.**

| Category | Symbol | Description |
|---|---|---|
| Client Model Structure | $C = \{c_1, \ldots, c_N\}$ | Set of all $N$ clients |
| | $C^{(r)} \subseteq C$ | Participating clients in round $r$ |
| | $T = \{T_1, \ldots, T_N\}$ | Set of tasks, one per client |
| | $T^{(r)} = \{T_i \mid c_i \in C^{(r)}\}$ | Set of participating tasks in round $r$ |
| Model Update | $\Delta W_{\text{enc}}^{(i)} \in \mathbb{R}^{D_{\text{enc}}}$ | Encoder update of client $i$ (derived by client model) |
| | $\Delta W_{\text{dec}}^{(i)} \in \mathbb{R}^{D_{\text{dec}}}$ | Decoder update of client $i$ (derived by client model) |
| | $\Delta W_{\text{enc}}' \in \mathbb{R}^{D_{\text{enc}}}$ | Aggregated encoder update from prototypes (derived by global model) |
| | $\Delta W_{\text{dec}}'^{(i)} \in \mathbb{R}^{D_{\text{dec}}}$ | Aggregated decoder update from prototypes of client $i$ (derived by global model) |
| Projection & Attention | $z_{\text{enc}}^{(i)} \in \mathbb{R}^d$ | Normalized encoder update of client $i$ (via projection of $W_{p,\text{enc}}$) |
| | $z_{\text{dec}}^{(i)} \in \mathbb{R}^d$ | Normalized decoder update of client $i$ (via projection of $W_{p,\text{dec}}$) |
| | $s_m^{z_{\text{enc}}^{(i)}} \in \mathbb{R}$ | Cosine similarity between $z_{\text{enc}}^{(i)}$ and prototype $P_m$ |
| | $\alpha_m^{z_{\text{enc}}^{(i)}} \in \mathbb{R}$ | Full attention weights from $z_{\text{enc}}^{(i)}$ to prototype $P_m$ |
| | $\tilde{\alpha}_m^{z_{\text{dec}}^{(i)}} \in \mathbb{R}$ | top-$k$ attention weight from $z_{\text{dec}}^{(i)}$ to prototype $P_m$ |
| | $r_{\text{enc}}^{(i)} \in \mathbb{R}^d$ | Intermediate representation of encoder update from prototypes for client $i$ |
| | $\tilde{r}_{\text{enc}} \in \mathbb{R}^d$ | Mean intermediate representation of encoder update from prototypes of all clients |
| | $W_{p,\text{enc}} \in \mathbb{R}^{d \times D_{\text{enc}}}$ | Encoder projection matrix |
| | $W_{p,\text{dec}} \in \mathbb{R}^{d \times D_{\text{dec}}}$ | Decoder projection matrix |
| Prototype | $P_m \in \mathbb{R}^d$ | $m$-th prototype vector |
| | $P \in \mathbb{R}^{M \times d}$ | Matrix of all $M$ prototype vectors |
| | $u_m \in \mathbb{R}$ | Usage score of prototype $P_m$ |
| | $\tilde{u}_m \in \mathbb{R}$ | Normalized usage score of $P_m$ |
| | $i_m \in \mathbb{R}$ | Isolation score of prototype $P_m$ |

## A.2 ADDITIONAL EXPERIMENTAL DETAIL

### A.2.1 TRAINING DETAILS

**Client Model Architectures.** The client model structure follows the same setup as that used in FedHCA$^2$ (Lu et al., 2024), ensuring consistency in the federated multi-task learning configuration. Each client model consists of a common-architecture encoder and a task-specific decoder. The encoder is implemented using a pretrained Swin-T transformer (Liu et al., 2021), which extracts multi-scale feature maps through four hierarchical stages. To inform the decoder of the current task, a task-specific vector is provided as input. This vector is processed by a small fully connected network with LeakyReLU activations to generate a task embedding. This task embedding is then used to condition the decoder during training, allowing it to adjust its processing for different tasks. The decoder consists of four upsampling stages, each incorporating skip connections from the corresponding encoder layer. To fully recover the input resolution, an additional upsampling step is applied at the end. The decoder then generates task-specific predictions through a final 1×1 convolution.

**Optimization and Loss Functions.** Model optimization is performed using the AdamW optimizer with an initial learning rate and weight decay of 1e-4. A cosine learning rate schedule is applied, beginning with a 5-round warm-up phase. Each client performs one local training epoch per communication round, a setting commonly used in prior work (Lu et al., 2024), with a batch size of 4. The loss function is selected based on the task. Cross-entropy loss is used for semantic segmentation and human parts segmentation. For saliency detection, surface normal estimation, and edge detection, we use balanced cross-entropy loss, L1 loss, and weighted binary cross-entropy loss, respectively. In the case of edge detection, the positive and negative pixel weights are set to 0.95 and 0.05 to reflect the

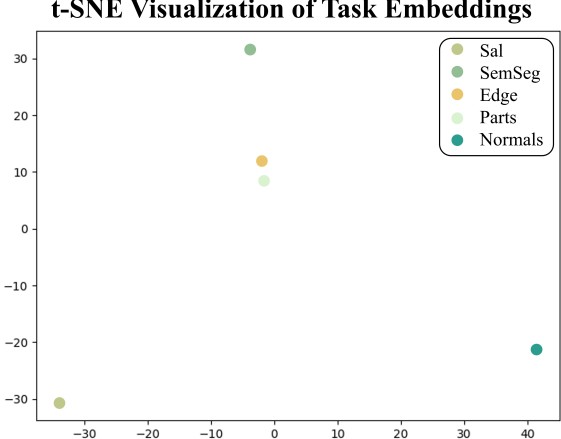

Figure 3: **t-SNE Visualization of Task Embeddings.** Distances were measured by projecting each task's decoder update into a shared PCA space and computing the L2 distance from the overall mean task representation. The results of distances are as follows: Sal (8.59), Edge (11.98), SemSeg (31.93), Parts (45.65), and Normals (46.46). Each point represents a task-specific decoder embedding, and colors indicate different tasks.

imbalance between edge and non-edge pixels. All experiments are implemented in PyTorch and conducted on four NVIDIA RTX-3090 GPUs.

**Data Augmentation.** We follow established data augmentation procedures widely used in previous studies (Kanakis et al., 2020; Ye and Xu, 2022; Maninis et al., 2019). Specifically, training images are randomly scaled with factors between 0.5 and 2.0, cropped to 512×512 resolution, horizontally flipped, and color-jittered. Image normalization is applied during both training and evaluation.

**Prototype Configuration.** We initialize the total number of prototypes to $M = 10$. The cumulative weight threshold for dynamic prototype selection in PGA is set to $\tau_k = 0.7$ and the prototype addition threshold in APA is $\delta_{add} = 0.3$. These values were empirically chosen as they showed the best performance in our sensitivity analysis, which is detailed in Section C.1. This configuration is used throughout all experiments unless stated otherwise.

**Baselines.** We follow previous studies (Lu et al., 2024; Yang et al., 2024) to adapt existing methods to the FMTL setting and compare our method with seven representative algorithms: FedAvg (McMahan et al., 2017), FedProx (Li et al., 2020), Ditto (Li et al., 2021), LANDER (Tran et al., 2024), FedTA (Yu et al., 2025), MOCHA (Smith et al., 2017), MaT-FL (Cai et al., 2023), FedHCA$^2$ (Lu et al., 2024). Following this FMTL setup, we further modify the setting to fit our FMTCL scenario by dynamically varying task participation across communication rounds. For FedAvg and FedProx, we aggregate the encoder and decoder parameters separately across clients. In Ditto, the encoder is shared globally while the decoder remains local to each client, and training is performed jointly with a regularization term for personalization. LANDER and FedTA also share the encoder but uses a task-specific latent representation to modulate the decoder. For MOCHA and MaT-FL, we adopt a parameter decoupling strategy, where only the encoder is shared and updated across clients, while the decoder remains local to each client. FedHCA$^2$ is evaluated by applying its original aggregation strategies for the encoder and decoder, adapted to the FMTL context.

A.2.2 Scenario Details

**Main Scenario.** To evaluate FedTAP under realistic task dynamics, we design a 50-round scenario involving five downstream tasks: semantic segmentation (SemSeg), human parts segmentation (Parts), saliency detection (Sal), surface normal estimation (Normals),

Table 8: **Task Participation Timelines.** Task participation in each round is determined by the assigned task state, with a check mark indicating presence and a cross mark indicating absence.

(a) 50 Rounds

| Round | Active | Inactive | Left | New |
|---|---|---|---|---|
| 1 - 5 | ✓ | ✓ | ✓ | ✗ |
| 6 - 10 | ✓ | ✓ | ✓ | ✗ |
| 11 - 15 | ✓ | ✓ | ✓ | ✗ |
| 16 - 20 | ✓ | ✗ | ✓ | ✗ |
| 21 - 25 | ✓ | ✗ | ✗ | ✗ |
| 26 - 30 | ✓ | ✗ | ✗ | ✗ |
| 31 - 35 | ✓ | ✗ | ✗ | ✓ |
| 36 - 40 | ✓ | ✓ | ✗ | ✓ |
| 41 - 45 | ✓ | ✓ | ✗ | ✓ |
| 46 - 50 | ✓ | ✓ | ✗ | ✓ |

(b) 100 Rounds

| Round | Active | Inactive | Left | New |
|---|---|---|---|---|
| 1 - 10 | ✓ | ✓ | ✓ | ✗ |
| 11 - 20 | ✓ | ✓ | ✓ | ✗ |
| 21 - 30 | ✓ | ✓ | ✓ | ✗ |
| 31 - 40 | ✓ | ✗ | ✓ | ✗ |
| 41 - 50 | ✓ | ✗ | ✗ | ✗ |
| 51 - 60 | ✓ | ✗ | ✗ | ✗ |
| 61 - 70 | ✓ | ✗ | ✗ | ✓ |
| 71 - 80 | ✓ | ✓ | ✗ | ✓ |
| 81 - 90 | ✓ | ✓ | ✗ | ✓ |
| 91 - 100 | ✓ | ✓ | ✗ | ✓ |

and edge detection (Edge). Each task is assigned one of four participation states—Active, Inactive, Left, or New—based on how independent its representation is from the global model. To determine task independence, we extract decoder updates for each task in the first communication round, project them into a shared PCA space, and compute the L2 distance from the average task representation. A t-SNE visualization of these representations is shown in, Figure 3. A larger L2 distance implies that the task depends more on specialized features and is less aligned with shared representations across tasks, making generalization more challenging for the global model. Based on these distances, the tasks are ranked in the following order of increasing independence: Sal, SemSeg, Edge, Parts, and Normals. We assign task participation states in correspondence with this ranking.

- Sal, being the closest to the mean representation, is assigned the active state and participates throughout all rounds.

- SemSeg and Edge are assigned the inactive state and are temporarily removed during the middle stage of training.

- Parts is designated as the left task and permanently exits the federation after the initial phase.

- Normals, the most independent task, is assigned the new state and joins the federation only in the later rounds.

This assignment ensures that tasks with more distant representations, hence more challenging for the global model to learn their task-specific features, participate less frequently, thereby increasing the difficulty of generalization. Additionally, to independently observe the impact of each participation state, we assign different transition timings for each task state. Specifically, Inactive tasks (SemSeg and Edge) are present during rounds 1–15 and 36–50 but absent from rounds 16–35. The Left task (Parts) participates from rounds 1–25 and then permanently exits. The New task (Normals) joins in round 31 and remains until the end. By varying the timing of each task's entry or exit, we prevent overlaps between state transitions, which enables independent analysis of how each participation state affects learning and forgetting. We refer to this setup as the *Main Scenario* as shown in Table 8a, and all experiments reported in the main paper are conducted based on this configuration. By default, all results are obtained under this scenario unless explicitly stated otherwise. We additionally extend this setup to a 100-round, as shown in Table 8b.

**Diverse Scenario.** Beyond the main scenario described above, we conduct further experiments to assess the robustness of FedTAP under more generalized task dynamics, referred

Table 9: **Task-to-State Combinations in Diverse Scenario.** Five task-to-state combinations are designed to evaluate the robustness of FedTAP, with the first combination corresponding to the main scenario.

| Task State | Active | Inactive | Left | New |
|---|---|---|---|---|
| Task Comb. 1 | Sal | Edge, SemSeg | Parts | Normals |
| Task Comb. 2 | Normals | Sal, Edge | SemSeg | Parts |
| Task Comb. 3 | Parts | Normals, Sal | Edge | SemSeg |
| Task Comb. 4 | SemSeg | Parts, Normals | Sal | Edge |
| Task Comb. 5 | Edge | SemSeg, Parts | Normals | Sal |

Table 10: **Comparison of Normalization Strategies.** The comparison shows that using L2 normalization for client updates in FedTAP results in higher performance than no normalization, Min-Max and L1 normalization.

| Norm. | Active | Inactive | Left | New | Avg. |
|---|---|---|---|---|---|
| No Norm. | 80.04 | 72.17 | 26.93 | 79.59 | 51.74 |
| Min-Max | 81.09 | 72.90 | 27.45 | 79.06 | 52.10 |
| L1 | 83.83 | 78.64 | 28.74 | 80.39 | 54.32 |
| **L2** | **86.12** | **84.24** | **33.90** | **82.75** | **57.40** |

to as the *Diverse Scenario*. To this end, we construct five different task-to-state combinations by rotating the participation states—Active, Inactive, Left, and New—across the five tasks—Sal, SemSeg, Edge, Parts, and Normals. In each configuration, one task is assigned to each state (with two tasks marked as Inactive), resulting in five unique mappings. For example, in one alternative configuration, Normals is Active, Sal and Edge are Inactive, SemSeg is Left, and Parts is New. The task-to-state combinations used in Diverse scenario are summarized in Table 9.

## B  ADDITIONAL ANALYSIS OF FEDTAP

### B.1  ABLATION OF PROTOTYPE-GUIDED AGGREGATION (PGA).

**Ablation of Normalization**    To investigate the effect of normalization strategies on representation learning within our method, we compare various normalization techniques applied to client updates, including L1, L2, min-max, and no normalization. As shown in Table 10, FedTAP, which adopt L2 normalization, achieves higher performance across all tasks. This performance gap can be attributed to the fundamental difference in how these normalization strategies affect vector geometry. Without normalization, client updates retain their original magnitudes, resulting in scale discrepancies across tasks. This leads to unstable representation alignment, where updates from semantically similar tasks may be treated unequally. Min-max normalization rescales values to a fixed range but does not preserve the directional information. While L1 normalization can induce sparsity, it distorts the original direction of client updates. Since FedTAP relies on prototype-based attention guided by cosine similarity, preserving directionality is crucial. Thus, this distortion leads to less reliable prototype selection and increased task interference. In contrast, L2 normalization preserves the directionality of updates, enabling more stable and semantically meaningful alignment between task representations. This directional consistency allows the prototype attention mechanism to generalize better across heterogeneous and temporally shifting tasks, resulting in more robust performance in FMTCL.

**Ablation of Aggregation Strategies.**    To validate the design principle of our asymmetric aggregation strategy for the encoder and decoder, we conducted an ablation study where we intentionally reversed the aggregation method for each component. For the encoder

Table 11: **Comparison of Aggregation Strategies for the Encoder.** The comparison shows that aggregating encoder updates using the full set of prototypes results in a higher average performance than using a selective subset.

| Encoder-side Agg. | Active | Inactive | Left | New | Avg. |
|---|---|---|---|---|---|
| Selective | 85.03 | 80.93 | 30.96 | 81.49 | 55.68 |
| **FedTAP (Ours)** | **86.12** | **84.24** | **33.90** | **82.75** | **57.40** |

Table 12: **Comparison of Aggregation Strategies for the Decoder.** The comparison shows that using a selective subset of prototypes for decoder aggregation achieves a higher average performance than aggregating over the full set.

| Decoder-side Agg. | Active | Inactive | Left | New | Avg. |
|---|---|---|---|---|---|
| Full | 83.90 | 78.85 | 29.54 | 81.09 | 54.67 |
| **FedTAP (Ours)** | **86.12** | **84.24** | **33.90** | **82.75** | **57.40** |

analysis, we compared our default 'Full' aggregation approach, which leverages the entire set of prototypes to learn generalizable features, against a 'Selective' approach that mimics the decoder by using only a subset of prototypes based on cosine similarity. As shown in Table 11, 'Full' aggregation method achieved significantly higher performance than the 'Selective' method. This result indicates that the encoder must aggregate over the entire prototype set to prevent it from becoming biased towards the features of a few specific tasks, thereby ensuring it develops a balanced and general-purpose representation beneficial for all clients. In contrast, for the decoder analysis, we compared our 'Selective' method, designed for task specialization, against a 'Full' method that aggregates updates over all prototypes. Table 12 shows that 'Selective' approach substantially outperformed the 'Full' aggregation method. This suggests that for the task-specific decoder, forcing the integration of information from all prototypes, including those irrelevant to its task, acts as noise and degrades performance. Taken together, these contrasting results provide strong evidence that our asymmetric design, by optimizing the aggregation strategy for the encoder's role of generalization and the decoder's role of specialization, effectively minimizes inter-task interference while maximizing the benefits of collaboration.

**Ablation on the Prototype Subset Selection.** To further investigate the decoder's selective aggregation, we evaluated the effectiveness of our dynamic selection mechanism. We compared our 'Dynamic' approach, which adaptively selects a variable number of prototypes based on a cumulative weight threshold $\tau_k$, against a 'Fixed' baseline that always selects a constant number of top-$k$ most similar prototypes. To ensure a fair comparison, the value of $k$ was set to the rounded integer average of the number of prototypes selected by our dynamic method over the entire training process. The result in Table 13 shows that the 'Dynamic' approach consistently outperforms the 'Fixed' approach across all task categories. This performance gap arises from the inherent inflexibility of the fixed-$k$ method, which proves suboptimal in two key scenarios. First, when an update is well-aligned with only a small subset of existing prototypes, a fixed-$k$ selection can be harmful because it forces the inclusion of less relevant prototypes, adding noise that distorts the update's true directional signal. Second, when an update requires a careful combination of multiple prototypes, such as those from a new task, the fixed-$k$ selection restricts the model's expressive capacity and prevents it from fully capturing the novel information in the update. Our dynamic mechanism addresses this limitation by flexibly adjusting the subset size to the representational needs of each individual update. This adaptability is essential for achieving both robustness and high performance in the challenging FMTCL setting.

**Ablation of Integration Strategies in Decoder-Side Aggregation.** To analyze the integration strategy for the decoder-side update, as detailed in Eq. (9), we compare our method, which combines the original client update with the prototype-guided representation, against a 'Prototype Attention Only' baseline that discards the original update. As shown in

Table 13: **Comparison of Dynamic and Fixed Prototype Selection.** The results show that the dynamic selection of a prototype subset achieves consistently higher performance across all task types than the selection of a fixed number of top-$k$ prototypes.

| $P$ Selection | Active | Inactive | Left | New | Avg. |
|---|---|---|---|---|---|
| Fixed | 84.15 | 80.96 | 30.36 | 30.31 | 55.15 |
| **Dynamic** | **86.12** | **84.24** | **33.90** | **82.75** | **57.40** |

Table 14: **Comparison of Integration Strategies in Decoder-Side Aggregation.** The results indicate that combining the original client update with the prototype-guided representation yields a higher average performance than using the prototype-guided representation alone, as described in Eq. (9).

| Integration | Active | Inactive | Left | New | Avg. |
|---|---|---|---|---|---|
| Prototype Attention Only | 85.11 | 81.59 | 31.89 | 81.53 | 56.02 |
| **FedTAP (Ours)** | **86.12** | **84.24** | **33.90** | **82.75** | **57.40** |

Table 14, our integration strategy shows better performance than the 'Prototype Attention Only' baseline. This result reveals a key insight for personalization in FL that optimal performance is achieved by combining the collaborative guidance of prototypes with the local signals of the original update.

## B.2 Analysis on the Learned Prototype Space.

**Results on Prototype Disentanglement and Alignment with Task Representation.** To examine how well FedTAP organizes the prototype space with respect to task representation, we visualize both the learned prototypes and task embeddings using t-SNE at the final round of training. As shown in Figure 4, the prototypes are well separated from one another without overlapping and positioned close to the regions associated with their respective tasks. Such structure is driven by the combined effect of the two loss functions in TPL. This indicates that the learned prototypes are both well disentangled and semantically aligned.

## C Generalization and Robustness of FedTAP

### C.1 Results on Hyperparameter Sensitivity

Figure 5 shows the sensitivity analysis of three key hyperparameters: the initial number of prototypes $M$, the dynamic selection threshold $\tau_k$ in PGA, and the prototype addition threshold $\delta_{add}$ in APA. Across all three parameters, FedTAP consistently outperforms FedHCA$^2$, the best-performing baseline, with the highest performance achieved at $M = 10$, $\tau_k = 0.7$ and $\delta_{add} = 0.3$.

### C.2 Results Under Various Number of Clients per Task.

To evaluate the scalability of our method, we varied the number of clients per task from 2 to 10 and compared the performance with FedHCA$^2$. As shown in Table 15, the performance of both methods degrades as the number of clients increases, as each client consequently holds fewer data samples for local training. However, FedTAP consistently perform higher than FedHCA$^2$ across all scales. This suggests that our prototype-based aggregation more effectively synthesizes knowledge even when individual client updates are derived from more limited local data.

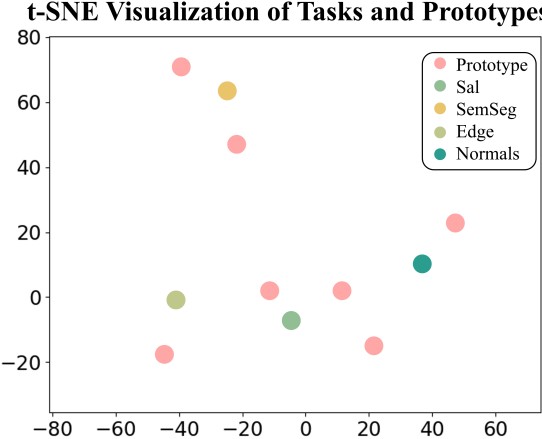

Figure 4: **t-SNE Visualization of Tasks and Prototypes.** The results show that prototypes are well disentangled and positioned to align with the characteristics of their associated tasks, as observed at the final round.

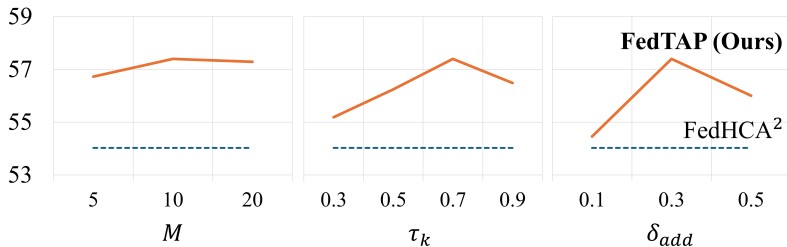

Figure 5: **Hyperparameter Sensitivity.** The results shows that FedTAP maintains stable performance across different values of $M$, $\tau_k$, and $\delta_{\text{add}}$, demonstrating robustness to hyperparameter variation.

Table 15: **Performance with a Varying Number of Clients per Task.** The comparison shows that while performance for both methods decreases with more clients per task, FedTAP consistently outperforms the FedHCA$^2$.

| Number of clients per task | Method | |
|:---:|:---:|:---:|
| | FedHCA$^2$ (Lu et al., 2024) | **FedTAP (Ours)** |
| 2 | 54.03 | **57.40** |
| 3 | 50.19 | **52.16** |
| 5 | 28.93 | **34.76** |
| 10 | 9.10 | **13.37** |

Table 16: **Comparison of Model Performance under Main and Diverse Scenarios.** The comparison highlights that FedTAP consistently outperforms baseline methods across the main scenario, its 100-round extension, and the five task-to-state combinations defined in the diverse scenario, demonstrating strong robustness to dynamic task compositions.

| Setting | Method | Scenario | Main | Main | Diverse |
|---|---|---|---|---|---|
| | | # of Rounds | 50 | 100 | 50 |
| | Local | | 51.82 | 51.88 | 51.95 |
| Traditional FL | FedAvg (McMahan et al., 2017) | | 52.19 | 52.19 | 52.30 |
| | FedProx (Li et al., 2020) | | 51.87 | 51.93 | 51.89 |
| | Ditto (Li et al., 2021) | | 52.86 | 52.89 | 52.85 |
| FCL | LANDER (Tran et al., 2024) | | 52.94 | 52.92 | 52.04 |
| | FedTA (Yu et al., 2025) | | 53.08 | 53.27 | 53.46 |
| FMTL | MOCHA (Smith et al., 2017) | | 53.24 | 53.31 | 53.54 |
| | MaT-FL (Cai et al., 2023) | | 53.71 | 54.01 | 53.98 |
| | FedHCA$^2$ (Lu et al., 2024) | | 54.03 | 54.35 | 54.14 |
| | **FedTAP (Ours)** | | **57.40** | **57.51** | **57.49** |

Table 17: **Comparison of FedTAP and baselines on the NYU Depth v2 dataset.** Results on NYU Depth v2 using the main scenario setup from PASCAL-Context as described in Table 8a, showing that FedTAP consistently outperforms all baselines across evaluation metrics.

| Setting | Method | Task | SemSeg (mIoU ↑) | Depth (RMSE ↓) | Normals (mErr ↓) | Edge (odsF ↑) |
|---|---|---|---|---|---|---|
| | | Task Status | Inactive | Left | New | Active |
| | Local | | 17.10 | 0.7978 | 23.19 | 55.03 |
| Traditional FL | FedAvg (McMahan et al., 2017) | | 17.52 | 0.7641 | 23.17 | 54.12 |
| | FedProx (Li et al., 2020) | | 18.49 | 0.7510 | 22.55 | 55.18 |
| | Ditto (Li et al., 2021) | | 18.04 | 0.7694 | 22.36 | 56.10 |
| FCL | LANDER (Tran et al., 2024) | | 18.91 | 0.7409 | 21.79 | 57.17 |
| | FedTA (Yu et al., 2025) | | 19.05 | 0.7357 | 21.20 | 57.69 |
| FMTL | MOCHA (Smith et al., 2017) | | 18.87 | 0.7258 | 22.18 | 57.03 |
| | MaT-FL (Cai et al., 2023) | | 19.01 | 0.7092 | 23.07 | 57.23 |
| | FedHCA$^2$ (Lu et al., 2024) | | 19.12 | 0.6915 | 22.21 | 57.81 |
| | **FedTAP (Ours)** | | **20.59** | **0.6491** | **20.94** | **58.43** |

## C.3 RESULTS UNDER VARIOUS SCENARIOS

We evaluate FedTAP under various task scenarios to assess its robustness. As shown in Table 16, FedTAP achieves the best performance not only in the original task combination used in the main scenario—where Sal is assigned as Active, SemSeg and Edge as Inactive, Parts as Left, and Normals as New—and when this scenario is extended to 100 rounds, but also on average across all five task-to-state combinations defined in the diverse scenario, as illustrated in Table 9. This demonstrates the robustness and generalization capability of FedTAP across a wide range of dynamic task composition settings.

## C.4 RESULTS ON ANOTHER DATASET

To evaluate the effectiveness of the proposed FedTAP method, we conducted experiments on the NYU Depth v2 dataset (Silberman et al., 2012), a widely used benchmark for multi-task learning (MTL), and compared its performance with several existing baselines. NYU Depth v2 consists of indoor scene images, and we followed the experimental configuration described in previous work (Lu et al., 2024). This dataset includes four tasks: edge detection, semantic segmentation, surface normal estimation, and depth estimation. The experiment adopts the

Table 18: **Computational Cost and Communication Overhead.** The result shows that FedTAP is the most efficient method, achieving the lowest FLOPs/Score and no additional communication overhead compared to FedAvg.

| Method | Training FLOPs/Round | Data Exchange/Round | FLOPs/Score($\downarrow$) |
|---|---|---|---|
| FedAvg (McMahan et al., 2017) | $1.0 \times$ ($\approx 1.13 \times 10^{13}$) | $1.0 \times$ ($\approx 166$MB $\uparrow\downarrow$ ) | $2.18 \times 10^{11}$ |
| FedHCA$^2$ (Lu et al., 2024) | $1.7 \times$ | $1.0 \times$ | $3.57 \times 10^{11}$ |
| **FedTAP (Ours)** | $1.1 \times$ | $1.0 \times$ | $2.12 \times 10^{11}$ |

same main scenario used in PASCAL-Context (Mottaghi et al., 2014), shown in Table 8a, which runs for a total of 50 rounds. The only difference is that the dataset includes a different set of tasks, resulting in a different assignment of task states. Each of the four task states—Active, Inactive, Left, and New—is assigned to exactly one task: semantic segmentation (SemSeg) is inactive, depth is left, normals is new, and edge is active. As in the PASCAL-Context setting, evaluation is based on the performance of the client model at the final round in which each task is involved. For example, SemSeg, which is inactive, is evaluated at round 50, while depth, designated as left, is evaluated at round 20. As shown in Table 17, FedTAP outperforms the baselines across all evaluation metrics on NYU Depth v2. This indicates that FedTAP achieves consistent and robust performance improvements not only on PASCAL-Context but also on other representative MTL benchmarks, demonstrating strong generalization ability and practical applicability across diverse settings.

# D    Efficiency of FedTAP

## D.1    Results on Computational Cost and Communication Overhead

We provide a detailed analysis of the computational cost and communication overhead, with results summarized in Table 18. All values are reported as ratios relative to the FedAvg baseline. The computational cost was calculated as the total FLOPs from all client-side training and the server-side aggregation for each round. The communication overhead, defined as Data Exchange per Round, quantifies the total data transmitted between clients and the server in each round. Since FedTAP only transmits the standard updates for the local encoder and decoder, not the prototypes themselves, it incurs no additional communication overhead. To evaluate the trade-off between computation and performance, we use the FLOPs per Score metric, where a lower value indicates higher efficiency. By achieving the lowest score, FedTAP demonstrates that it is the most efficient method.

