# OpenReview forum: "FedTAP: Federated Multi-Task Continual Learning via Dynamic Task-Aware Prototypes"
_ICLR.cc/2026/Conference — ICLR 2026 Conference Withdrawn Submission_

### Official Review · Reviewer_xFpe · 2025-10-15

**Soundness:** 2
**Presentation:** 2
**Contribution:** 3
**Rating:** 4
**Confidence:** 4

**Summary:**

This paper introduces FMTCL, which handles task heterogeneity, temporal data shifts, and dynamic task participation. The proposed method, FedTAP, addresses these challenges through (i) Prototype-Guided Aggregation, (ii) Task-Aware Prototype Learning, and (iii) Adaptive Prototype Allocation. Experiments on multi-task benchmarks demonstrate state-of-the-art performance and adaptability in dynamic environments.

**Strengths:**

- Data heterogeneity is a practical problem, reflecting real-world data heterogeneity.

- Table 3 shows the effectiveness of the proposed components.

**Weaknesses:**

- Figure 1 should be simplified and refined. The current version is overly complex, making it difficult to grasp the overall idea of the proposed method.

- Writing and structure need improvement. For instance, prototypes are introduced in Sec. 3.2 without being clearly defined; their meaning only becomes clear after reading Sec. 3.3. Overall, the paper’s structure should be reorganized to improve readability and logical flow.

- Heavy reliance on hyperparameters. As shown in the hyperparameter analysis (Appendix), the method appears highly sensitive to hyperparameter choices. Although the paper uses unified hyperparameters across experiments, such sensitivity could hinder real-world applicability, where task heterogeneity and distribution shifts vary significantly.

- Insufficient baselines: In the FMTCL setup, comparisons with FMTL and FCL baselines are unfair comparisons. The experimental results (e.g., Table 2) should include a broader range of baselines that combine both FCL and FMTL methods to more clearly demonstrate the effectiveness of the proposed approach under fair comparisons.

- Marginal improvements in Table 17: While the proposed method shows significant gains on the PASCAL-Context benchmark, its improvements on NYU Depth v2 are only marginal. The paper should provide a detailed analysis or discussion explaining why the method yields limited improvements on the NYU Depth v2 dataset.

**Questions:**

- Performance in other setups: It would be helpful to evaluate whether the proposed method also performs well in FMTL and FCL settings to show its generalizability. For example, whether it achieves higher accuracy and maintains a stable number of prototypes under consistent data distributions.

---

### Official Review · Reviewer_3n5p · 2025-10-17

**Soundness:** 1
**Presentation:** 3
**Contribution:** 2
**Rating:** 2
**Confidence:** 5

**Summary:**

The paper addresses continual learning challenges in federated multi-task systems. To tackle these dual issues, the authors propose FedTAP, a prototype-based aggregation mechanism that helps the server perform more accurate model aggregation.

**Strengths:**

1. The paper tackles a highly challenging problem that combines federated learning, continual learning, and multi-task learning. Consequently, both the datasets and model architectures used in the study are complex and demanding.
2. The paper has good representation and organization.

**Weaknesses:**

1. Although the continual learning (CL) are mentioned as the main contribution of the paper, the characteristics of CL is unlikely to be seen among the paper, which make the paper lack of coherence and contributions of the paper seems to be limited. To be more specific,
   - The concept of catastrophic forgetting, a core challenge in CL, is only briefly mentioned. While the authors claim that their method is robust to CL within a federated multi-task learning (FMTL) framework, they fail to explain or justify how the proposed algorithm mitigates catastrophic forgetting. Moreover, the key challenges arising from the intersection of CL and MTL are not clearly identified, leaving unclear why existing CL or MTL methods cannot address this joint problem.
   - The use of PASCAL-Context and NYU Depth v2 datasets is mentioned, yet the paper does not clearly describe how the tasks are split to simulate the FCL scenario.
   - This lack of clarity extends to the experimental results, where the performance of traditional FL, FCL, and FMTL settings appears nearly identical across tasks. Such results raise doubts about whether FedTAP was truly evaluated under a continual learning framework.
2. The literature review on FCL is incomplete and requires substantial improvement to provide a more comprehensive understanding of prior work.
3. The baselines used in the paper could be significantly improved. Except for LANDER and FedTA, most compared methods are not FCL algorithms. Moreover, it is unclear how FedTA and LANDER were adapted to the multi-task setting. Providing more clarification would greatly enhance the reproducibility of the benchmarking.
4. The method description lacks clarity and detailed explanation. Several key components (e.g., Eqs. (3)-(9)) are introduced without sufficient justification, making it difficult to assess the novelty of the contribution. The prototype aggregation in Eq. (5) appears to be an incremental variation of existing server-side aggregation methods. The gradient computations in Eqs. (6) and (9) are unconventional and not supported by concrete theoretical foundations. Furthermore, Eq. (8) does not clearly show the relationship between the objective function and its constraints.
5. The prototype learning framework deviates from standard prototypical network practices. While Eq. (10) is reasonable, Eq. (11) is introduced without sufficient discussion. Additionally, the paper claims that prototype learning involves only two losses as in Eq. (12), yet the main prototypical loss is missing. This cut is critical because prototype computation in Eq. (14) depends on well-separated clusters; otherwise, the learned prototypes may be unstable. Moreover, the Gaussian noise term in Eq. (14) is questionable, as it is representation-agnostic and likely to vanish after averaging due to the central limit theorem.
6. The prototype pruning mechanism needs stronger theoretical grounding and a clearer explanation of its principles and impact.
7. Many experimental evaluations provide limited insights. For instance, Fig. 3 merely presents scattered points, which the authors describe as prototypes, without offering deeper analysis or interpretation.
8. The evaluation protocol and metrics used for assessing FCL performance are insufficient, which limits the strength and validity of the paper’s conclusions.
9. Since the paper claims to reduce issues by exchanging prototypes within the system, it is essential to evaluate the communication cost to validate the proposed method’s efficiency and practicality.

**Questions:**

See Weaknesses

---

### Official Review · Reviewer_RAmi · 2025-10-27

**Soundness:** 1
**Presentation:** 2
**Contribution:** 1
**Rating:** 2
**Confidence:** 5

**Summary:**

The paper proposes FedTAP, a novel framework for Federated Multi-Task Continual Learning (FMTCL) that combines task heterogeneity, temporal data shifts, and dynamic task participation. Instead of directly aggregating model parameters, FedTAP introduces a prototype-based representation space to guide global updates with three key components. FedTAP effectively mitigates catastrophic forgetting, adapts to evolving task compositions, and achieves state-of-the-art performance across multiple datasets.

**Strengths:**

- The paper is well-written and easy to follow.
- The experimental results are extensive and convincing.

**Weaknesses:**

- It is clear that the font is incorrect and aesthetically unappealing, which could potentially lead to a desk rejection of the paper.
- The authors’ understanding of FCL appears to be wrong. In Line 88, they claim that FCL cannot use separate models, arguing that this would reduce shared knowledge between different tasks. However, in FCL, the continual arrival of new tasks naturally leads to catastrophic forgetting of previous knowledge. Using separate models to isolate task-specific knowledge, often combined with regularization-based techniques to facilitate knowledge transfer and share similar representations, is a well-established strategy in the field. Even more concerning, in Line 154, the authors state that existing FCL works assume a static class set. In fact, the very papers they cite explicitly adopt the opposite assumption that new classes continuously arrive. This raises doubts about the authors’ fundamental understanding of FCL and the accuracy of their literature review.
- Based on the FMTCL challenge proposed by the authors, I find that FedTAP itself does not introduce any fundamentally new techniques or insights. Using prototypes to separate different tasks is also a common strategy in Personalized FL, dating back four to five years. The authors merely adopt this existing idea without providing any new insight into how it could be integrated with the proposed FMTCL framework. Moreover, applying prototypes in federated continual learning (FCL) has already been explored extensively in prior works, such as Pilora and FedProk. The paper does not clearly articulate how its approach differs from these existing studies. Overall, the work feels more like an engineering-oriented technical report rather than a paper offering substantial methodological innovation.
- In terms of experimental setup, the authors should have included FCL-specific baselines, as catastrophic forgetting has a significant impact on model performance in continual learning. Without appropriate anti-forgetting mechanisms, models typically forget almost all knowledge from previous tasks. However, the results presented in the paper show that the proposed FMTL methods consistently outperform FCL methods, which is quite puzzling. This may indicate that the experimental setup is overly simplified and biased toward FMTL settings, whereas the main focus of the paper is supposedly on FCL. In addition, the number of FCL baselines is insufficient. The field of FCL has seen substantial progress over the past three years, including works on dynamic architectures and generative replay-based methods, none of which are discussed or compared in this paper.

**Questions:**

Please refer to Weaknesses.

---

### Official Review · Reviewer_LWsg · 2025-10-30

**Soundness:** 2
**Presentation:** 3
**Contribution:** 3
**Rating:** 4
**Confidence:** 4

**Summary:**

This paper introduces Federated Multi-Task Continual Learning (FMTCL), a novel and highly practical FL scenario that simultaneously addresses three interconnected challenges: task heterogeneity (clients perform different tasks), temporal data shift (a client's data distribution changes over time), and dynamic task composition (tasks dynamically join and leave the federation).

To address the challenges in FMTCL setting, the authors propose FedTAP (Federated Task-Aware Prototype). The core innovation is a shift from direct parameter aggregation to an indirect, prototype-based aggregation in a shared representation space. This design elegantly resolves the fundamental conflict between adaptation (needed for new tasks) and stability (needed to retain old knowledge) that plagues naive combinations of FMTL and FCL.

The paper provides extensive experiments on standard multi-task benchmarks under diverse dynamic scenarios, convincingly demonstrating state-of-the-art performance against strong baselines from FL, FCL, and FMTL.

**Strengths:**

1. Prototype-Guided Aggregation (PGA): This is the core aggregation mechanism. Instead of averaging model parameters, client updates (for both encoder and decoder) are projected into a shared prototype space. The aggregation is performed by representing each update as a combination of shared prototypes. Crucially, the design is asymmetric: encoder updates use all prototypes to learn general features, while decoder updates use a sparse, dynamic subset for task specialization. This effectively minimizes interference between heterogeneous tasks.

2. Task-Aware Prototype Learning (TPL): This component ensures the prototype pool is effective. It trains a diverse set of prototypes using two complementary losses: a Prototype Diversity (PD) loss that enforces orthogonality to reduce redundancy, and a Sparse Attention (SA) loss that encourages each task to focus on a compact set of relevant prototypes. This ensures the prototype space is expressive and task-aware.

3. Adaptive Prototype Allocation (APA): This component enables adaptation to dynamic task composition. It dynamically manages the prototype pool by adding new prototypes when existing ones cannot represent a client's update (indicating a novel task) and pruning prototypes that are both rarely used and semantically isolated. This prevents representational capacity misallocation and efficiently allocates resources to active tasks.

4. Efficiency: Notably, FedTAP achieves superior performance without increasing communication overhead compared to FedAvg, as only standard model updates are transmitted, not the prototypes.

**Weaknesses:**

1. Wrong claim about Federated Continual Learning. The claim that FCL "only considers temporal data shifts within a single-task" overlooks the well-established subfield of class-incremental FCL, where new tasks continually arrive. This misrepresentation weakens the foundational justification for the proposed FMTCL scenario, as it fails to acknowledge that dynamic task composition, albeit often within a unified paradigm like image classification, is already a core challenge studied in FCL. The authors should revise this statement to more accurately reflect the scope of FCL literature and more clearly delineate how their "task" concept differs from the "tasks" in class-incremental settings, such as through a higher degree of task heterogeneity.

2. Furthermore, the statement that 'FCL enforces stability, a model's ability to preserve knowledge over time, by constraining important parameters' is inaccurate because it is an overgeneralization. Constraining important parameters is merely one of several methodologies for achieving stability in FCL, and cannot be taken as the universal definition or sole approach.

3. The paper's explanation of the three core challenges "Task Heterogeneity, Temporal Data Shift, and Dynamic Task Composition" is insufficiently clear. As addressing these three challenges collectively is a central contribution of the proposed FMTCL scenario, the authors need to provide a more thorough exposition.

     To illustrate, the term 'Task heterogeneity' can encompass many meanings. Within a client's internal task sequences, it manifests as temporal task heterogeneity. Furthermore, in Federated Learning (FL), different clients may also be focused on distinct tasks, which constitutes another type of task heterogeneity. This combined spatial-temporal task heterogeneity is what FCL aims to address. However, the task heterogeneity mentioned by the authors in the paper seems more aligned with an extreme case—such as Client A learning semantic segmentation while another client performs saliency detection. The authors need to clarify this definition further. Moreover, I believe the practical significance of this type of extreme task heterogeneity is limited, as the core objectives of each client are too fundamentally different to form a cohesive FL system.

4. Insufficient Motivation. The logical progression and motivation leading to this specific design are not sufficiently articulated. The manuscript would significantly benefit from a more thorough "story-telling" approach that bridges the problem analysis and the proposed solution.

5. The experimental setup lacks clarity. The authors should meticulously detail the data and task allocation within the experimental section to clearly highlight how this setting simultaneously embodies Task Heterogeneity, Temporal Data Shift, and Dynamic Task Composition.

**Questions:**

1. How do the pre-defined threshold determined?

2. How does the learnable projection matrix perform its learning? Is this mapping matrix susceptible to catastrophic forgetting?

3. Generally, prototypes refer to the feature level of sample outputs. However, are the prototypes here related to the decoder's updates, and how is their dimensionality determined?

Please refer to Weaknesses for other questions.

---

### Note · Authors · 2025-11-14

I have read and agree with the venue's withdrawal policy on behalf of myself and my co-authors.